# Barriers and Facilitators to Nut Consumption: A Narrative Review

**DOI:** 10.3390/ijerph17239127

**Published:** 2020-12-07

**Authors:** Elizabeth P. Neale, Georgie Tran, Rachel C. Brown

**Affiliations:** 1School of Medicine, University of Wollongong, Wollongong, NSW 2522, Australia; gtran@uow.edu.au; 2Illawarra Health and Medical Research Institute, University of Wollongong, Wollongong, NSW 2522, Australia; 3Department of Human Nutrition, University of Otago, Dunedin 9016, Otago, New Zealand; rachel.brown@otago.ac.nz

**Keywords:** nut consumption, barriers, facilitators, dietary guidelines

## Abstract

Habitual nut intake is associated with a range of health benefits; however, population consumption data suggests that most individuals do not meet current recommendations for nut intake. The literature has highlighted a range of barriers and facilitators to nut consumption, which should be considered when designing strategies to promote nut intake. Common barriers include confusion regarding the effects of nut consumption on body weight, perceptions that nuts are high in fat, or too expensive, and challenges due to dentition issues or nut allergies. Conversely, demographic characteristics such as higher education and income level, and a healthier lifestyle overall, are associated with higher nut intakes. Health professionals appear to play an important role in promoting nut intake; however, research suggests that knowledge of the benefits of nut consumption could be improved in many health professions. Future strategies to increase nut intake to meet public health recommendations must clarify misconceptions of the specific benefits of nut consumption, specifically targeting sectors of the population known to have lower nut consumption, and educate health professionals to promote nut intake. In addition, given the relatively small body of evidence exploring barriers and facilitators to nut consumption, further research exploring these factors is justified.

## 1. Health Benefits of Nut Consumption

Nuts (including tree nuts and peanuts) are considered to be a core food which is recommended to be consumed as part of a healthy diet [1,2,3]. Habitual nut consumption has been associated with a range of health benefits, particularly in relation to a reduction in the risk of chronic diseases. For example, systematic reviews of observational evidence have demonstrated regular consumption of nuts to be associated with significant reductions in the risk of coronary heart disease and cardiovascular disease [4,5,6]. Nut intake has also been associated with a reduced risk of overweight and obesity [7]. Trends for reductions in risk of stroke, type 2 diabetes mellitus, and cancer have also been observed. However, significant reductions in the risk of these conditions have been less consistently reported [4,5,8,9,10].

Findings from observational evidence have also been demonstrated in interventions assessing the impact of nut consumption on health outcomes. One of the most well-known dietary intervention studies involving nut intake is the PREDIMED study [11], which explored the impact of a Mediterranean diet supplemented with either 30 g of nuts (consisting of 15 g of walnuts, 7.5 g of hazelnuts, and 7.5 g of almonds) or olive oil in a sample of 7447 individuals at high risk of cardiovascular disease. Compared to the control diet (which was designed to be low fat, however, fat intakes remained similar to a Mediterranean eating pattern), consumption of a Mediterranean diet incorporating nuts resulted in a 28% reduction in the composite risk of myocardial infarction, stroke, or death from a cardiovascular cause. Systematic reviews of dietary interventions have also demonstrated that provision of nuts results in improvements in total cholesterol, low-density lipoprotein cholesterol, and triglycerides [12,13], with stronger effects observed in studies assessing consumption of ≥60 g nuts/day [12].

In addition to the well-known benefits of nut consumption on cardiovascular disease, there is increasing evidence of the impact of nut intake on risk factors for other health outcomes. In a meta-analysis of randomised controlled trials, nut consumption resulted in significant improvements in flow-mediated dilation [14], a research tool used to measure endothelial function [15]. Inclusion of nuts within a diet may also lead to improvements in chronic inflammation, a risk factor for chronic diseases such as cardiovascular disease, although inconsistent results have been reported [14,16], highlighting this as an area requiring further exploration. In addition, there is now emerging research to suggest that consumption of nuts may result in favourable changes to the gut microbiome [17,18,19,20]. Conversely, meta-analyses suggest that nut intake does not impact blood pressure [12,21], although there is some evidence to suggest an effect may be present for specific nut types [22].

Nuts are a nutrient-dense food which are a rich source of protein, unsaturated fatty acids, fibre, and micronutrients such as folate, vitamin E, and magnesium [23,24]. Despite their energy-dense nature (due to their high fat content), nut consumption has had no observed adverse effect on body weight [21,25,26], with some evidence of small favourable effects on adiposity [27], particularly when nuts are substituted for other foods in the diet [26]. There are a number of mechanisms which may explain the lack of an effect of nut consumption on body weight. For instance, research conducted in almonds has shown increased excretion of fat in the faeces following almond consumption [28,29]. This finding is thought to be due in part to the structure of the nuts, where the fat is held in plant cell walls. As a result, the metabolisable energy available from nuts including almonds, walnuts, cashews, and pistachios is proposed to be 5–30% lower than estimated by Atwater factors [30,31,32,33]. Another purported mechanism is enhanced satiety with subsequent dietary compensation, due to the fibre and protein content of nuts [34]. Finally, emerging evidence suggests that nut consumption may play a role in supporting a healthy diet. Provision of nuts has been found to increase consumption of other core foods, and subsequently improve overall diet quality [14,35]. Modelling of the effect of replacing snacks with nuts resulted in diets which were significantly lower in added sugars, solid fats, saturated fat, and sodium, whilst being higher in unsaturated fats, fibre, and magnesium [36], highlighting the role of nut consumption in supporting a healthy overall dietary pattern.

Habitual consumption of nuts is, therefore, associated with a range of health benefits particularly reductions in the risk of cardiovascular and coronary heart disease, and improvements in biomarkers such as total and low-density lipoprotein cholesterol. Despite these benefits, there are potential disadvantages which should be considered with regard to nut consumption. Nuts may be consumed as salted varieties, and increased sodium intake is associated with increased blood pressure [37]. However, a small number of studies have compared the effect of different forms of nuts on health outcomes and found no significant differences in systolic and diastolic blood pressure after consumption of salted and unsalted nut varieties [38,39,40]. Based on these findings, Tey et al. [39] concluded that there was no adverse effect on blood pressure when consuming ≤ 285 mg/day sodium from nuts. In addition, it should be noted that a small proportion of the population experience peanut and tree nut allergies [41,42,43]. For these individuals, consuming nuts results in potentially life-threatening allergic reactions, requiring the complete avoidance of the offending nut type in the diet.

## 2. Guidelines for Nut Consumption

As a result of the health benefits associated with habitual nut consumption, nut intake is promoted in many food-based dietary guidelines globally [44]. However, there is some variation in the classification of nuts and specific recommendations made. Across guidelines, nuts are classified within different food groups or categories, including protein-rich foods such as meat and alternatives [1,2,3,45,46,47,48], fats and oils [49,50,51], or appear in their own food category [52,53]. Recommendations for nut consumption are typically based on consumption of these broader categories, for example the 2015–2020 Dietary Guidelines for Americans [2] recommends consuming 5 oz equivalents of nuts, seeds, and soy products per week as part of a 2000 kcal/day Healthy U.S.-Style Eating Pattern. Whilst there is some variation in the serving size recommended for nut consumption, serving sizes used in food-based dietary guidelines globally typically range from 15 to 30 g of nuts, aligning with the body of evidence for the benefits of nut intake [44].

## 3. Patterns of Nut Consumption

Despite the known health benefits associated with nut consumption, and the promotion of nut intake in many dietary guidelines globally, population nut intakes remains below recommended levels. For example, the results of the Global Burden of Disease Study 2017 considered an optimal intake of nuts and seeds to be 21 g per day, but found that global consumption was only 12% of this recommended level [54]. As a result, low consumption of nuts and seeds was found to be one of the leading risk factors for death and disability adjusted life years in many countries. In 2019, the Eat-Lancet Commission published their recommendations for a universal healthy reference diet, aimed at reducing mortality globally [55]. Nuts were included as a key component of the Eat-Lancet healthy reference diet, which recommended consumption of 50 g of nuts (which can include tree nuts and peanuts) per day (for an intake of 2500 kcal/day) as an alternative to red meat, with the authors specifically highlighting the need to substantially increase nut consumption globally to reach these intake levels.

National surveys provide insights into current nut intake around the globe, further highlighting the disconnect between recommendations for nut consumption and current intake levels. For example, in the United States of America, an analysis of nut and seed intake in adults from multiple cycles of the National Health and Nutrition Examination Survey (NHANES) from 1999 to 2012, found that while intake of nuts and seeds increased during this period, by 2012 only one-third of adults met dietary recommendations to consume one ounce of nuts and seeds (28.3 g) at least five times a week [56]. However, these results may be impacted by the inclusion of seeds in this analysis. Exploration of usual intake of tree nuts specifically in the NHANES 2005–2010 identified that only 6.8% of the population consumed at least ¼ ounces (7.1 g) of tree nuts per day [57,58]. While the mean usual intake of 44.3 g per day among nut consumers suggested that these individuals met recommendations, mean consumption of nuts overall was 3.3 g per day, highlighting low intakes of nuts in the population overall.

Similar patterns of nut consumption have been reported worldwide. Analysis of the European Prospective Investigation into Cancer and Nutrition (EPIC) study conducted in 10 European countries found that some variation in nut intake was present between individual countries (with lower intakes in Northern European countries compared to Southern European countries), with only 27.3% reporting consuming nuts from any source on the day of the 24 h recall [59]. Similarly, analysis of the 2011–2013 National Nutrition and Physical Activity Survey, a component of the nationally representative Australian Health Survey, found that less than 40% of Australians reported consuming any nuts (including nuts in mixed dishes such as breakfast cereals or muesli bars) [60]. Similar to the findings from the United States, mean nut intake in the broader Australian population was 4.6 g per day. However, when analyses were restricted to only those individuals who consumed nuts, mean intakes were 11.75 g per day, with only 5.6% of nut consumers meeting the recommendation of 30 g of nuts per day. Similar results were found for the United Kingdom, where analysis of the National Diet and Nutrition Survey (NDNS) 2008–2014 identified that median intake of nuts among nut consumers was 6.5 g per day [61]. Data on nut consumption patterns in New Zealand is similar to those reported in other countries [62]. Analysis of the 2008/09 New Zealand Adult Nutrition Survey found 28.9% of participants reported consuming any nut source (whole nuts, nut butter, hidden sources, e.g., nuts included in mixed dishes, breakfast cereals) on the day of the 24 h recall, with mean intakes of 5.2 g per day. When considering nut consumers only, intakes were higher at 17.9 g per day. Only 6.9% of participants reported consuming whole nuts on the day of the 24 h recall, with relatively high mean intakes of 40.3 g per day.

## 4. Barriers to Nut Consumption

Given the apparent discrepancy between population health recommendations for nut consumption, and the current levels of nut intake observed worldwide, it is prudent to consider factors which may act as barriers or facilitators to nut consumption. The literature has highlighted a number of factors which may impede nut consumption. Despite research suggesting that nut consumption does not adversely affect body weight, evidence has suggested that consumers may be confused about the effects of nuts on weight, due to concerns regarding consuming too much fat when nuts are regularly included in the diet [63]. For instance, we conducted an online survey with 71 individuals exploring perceptions of nut consumption among Australian consumers, which identified that 69% of consumers agreed that if eating nuts would cause weight gain, then that would prevent consumption of nuts [64]. In addition, when asked across multiple questions, approximately 15–25% of consumers agreed that consuming nuts would result in weight gain. Similarly, a survey of 710 individuals in New Zealand identified that the perceived effect on weight was a barrier to regular nut consumption among consumers [65]. Concerns regarding the impact of nut consumption on body weight have also been reported in consumers from the United States of America. Pawlak et al. conducted cross-sectional surveys with 124 individuals from a low socio-economic background [66] and 85 individuals with or at risk of cardiovascular disease and/or type 2 diabetes [67]. Of concern, 87% of individuals with or at risk of cardiovascular disease and/or type 2 diabetes and 37% of individuals from a low socio-economic background believed that eating nuts would cause weight gain. It is, therefore, evident that the perceived impact of nut consumption on body weight, which does not reflect the empirical evidence on the relationship between nut intake and body weight, is a barrier to increased intake. Considering the significance of this factor as a barrier to regular nut consumption, it is important that confusion regarding the effects of nut consumption on weight gain among consumers is resolved. Consumer education on the effects of nut intake on body weight should be addressed when developing future strategies to support nut consumption.

In addition, concerns regarding the fat content of nuts and nut butters have been reported in the literature, and may be an underlying reason why confusion exists regarding the effects of nut intake on body weight. The cross-sectional survey conducted in New Zealand identified that one of the top five reasons selected by respondents who avoided nut butters was that they were perceived to be unhealthy and high in fat [65]. This was similar to the findings from the Australian survey [64], where 62% of participants agreed that if eating nuts would cause them to eat too much fat, then that would prevent them from eating nuts or nut butters. Conversely, almost half of the participants agreed that if nuts were lower in fat, then that would increase their consumption of nuts or nut butters, and 70% of participants agreed that if eating nuts would help them get the right balance of good fats, then that would also increase their consumption of nuts or nut butters. Given that nuts are a good source of unsaturated fatty acids, there may be opportunity to educate consumers on the favourable fat profile of nuts to overcome this perceived barrier to consumption.

In addition to concerns regarding the fat content and impact on body weight, there are also other common barriers to nut consumption to note. A survey of 710 members of the general public in New Zealand found that the primary deterrent to increased nut consumption was if their cost was too high [63]. Similarly, the Australian survey with 71 individuals [64] identified that 64% of these consumers agreed that if nuts were more affordable they would increase their consumption of nuts or nut butters. Similarly, 65% agreed that if eating nuts would cost too much money, then that would prevent them from eating nuts or nut butters. It should be noted that participants in this survey tended to have a high income compared to the median Australian income, meaning that financial barriers to nut consumption observed in this group of participants may not reflect the rest of the population. The observations may therefore be amplified if a study was to be done on a larger random sample of participants. Price was also found to be inversely associated with the intention to consume nuts in a survey of 451 consumers from Zhejiang Provence, China [68], and highlighted as a barrier to regular nut consumption in a survey of 124 consumers from the United States of America [66]. While nuts may be less costly than other snacks such as potato chips and cereal bars [63], concerns regarding the price of nuts are evidently a common barrier to their consumption.

Dentition issues can make eating nuts inconvenient and uncomfortable, and therefore, may act as a possible barrier to nut consumption. An online survey of 204 Australian health professionals found that over half of the participants reported that their clients stated they had dental issues, making it inconvenient and uncomfortable to consume nuts [64]. This finding was similar to other countries such as New Zealand where dentition was a commonly reported barrier to nut consumption [65]. The New Zealand study also found that nut avoiders were more likely to avoid eating nuts because of dental issues, compared with nut butter avoiders. With evidence suggesting that there are no significant differences in health benefits between consuming different forms of nuts, including nut butters [69,70], it is reasonable that alternative forms of nuts that may pose fewer challenges for those with poor dentition could be recommended to this group of consumers.

Finally, a common nut consumption barrier is the presence of an allergy to nuts. A sample of Australian health professionals reported that the presence of a nut allergy was a common barrier to nut intake reported by their clients [64]. Similar findings were observed in New Zealand, where 15% of participants who did not consume nuts or nut butters reported that it was because they were allergic to nuts [65]. Additionally, 8% of participants did not consume nuts or nut butters because they lived with or were in close contact with someone who was allergic to nuts. Interestingly, these values are higher than the prevalence of peanut and tree nut allergy (confirmed using food challenges) reported in the literature, which is estimated to range from less than 1% to approximately 4% [41,42,43]. The variation between the proportion of individuals who report allergies as a barrier to nut intake and estimations of allergy prevalence may be the result of inaccurate self-reporting of nut allergies. Regardless, although a nut allergy may present an insurmountable barrier to nut consumption, it is still important to note this as a common barrier to regular nut consumption, to gain a deeper understanding of all the reasons why nut consumption levels might be poor. In addition, given that research has highlighted the importance of early introduction of peanuts in infants at high risk of peanut allergies [71], further research exploring whether individuals without diagnosed allergies are unnecessarily avoiding these foods is warranted.

## 5. Facilitators of Nut Consumption

Observational evidence has highlighted key characteristics which tend to be associated with higher consumption of nuts. For instance, a higher level of education or socio-economic status [57,60,62] and factors associated with a healthier lifestyle overall such as non-smoking [57,61] and higher levels of physical activity [57] have been associated with increased nut intake. In addition, increased age appears to be associated with higher nut consumption [57,61], although there is some evidence that this may peak in young and middle aged adults, as opposed to older adults [60,62]. Conversely, intakes of nut butters may be higher among younger age groups [62]. Identification of these factors provide insights into the population groups who may be less likely to meet recommendations for nut consumption, which can be valuable when designing targeted public health or dietetic strategies to increase nut intake.

While characteristics associated with nut consumption have been consistently demonstrated in the literature, there remains a paucity of research exploring behavioural factors or perceptions which may facilitate nut consumption. Pawlak et al. [66] conducted a cross-sectional survey with 124 individuals to explore attitudes, beliefs, and barriers towards regularly consuming nuts in a low socio-economic cohort in the United States of America. While nuts were generally considered to be healthy and high in protein, only one-third of participants believed that eating nuts may help to reduce cholesterol levels. Similar results were found when beliefs, attitudes, barriers and benefits, and knowledge of nut consumption was explored in a sample of 85 individuals with either diagnosed cardiovascular disease and/or type 2 diabetes, or who were at risk of these conditions [67]. Approximately half of the participants consumed nuts once to twice a week, with three-quarters of participants agreeing or strongly agreeing that nuts were healthy and high in nutrients. Similarly, two-thirds of participants agreed that peanuts and walnuts were a good source of omega-3 fatty acids. However, in contrast to these results, over half of the participants were either unaware or disagreed that nut consumption had a beneficial impact on cholesterol levels. As previously outlined, concerns regarding the effect of nuts on body weight were also evident in both surveys. Taken together, these results suggest that while the individuals surveyed largely agreed with the concept of nuts as a healthy food, they were not familiar with the specific health benefits associated with nut intake, with concerns relating to the impact of nut intake on body weight remaining.

Further insights into factors which may support nut consumption have been provided by cross-sectional surveys of consumers. Investigation into knowledge and perceptions of nut intake in 710 randomly selected New Zealand adults identified that the most commonly reported reason for choosing to consume nuts or nut butters was an enjoyment of the taste of nuts [65]. It should be noted, however, that when compared with reported consumption of nuts, enjoying the taste of nuts was not significantly associated with consumption levels. Other common reasons for eating nuts reported by participants in this survey included that nuts were healthy or nutritious, convenient and portable, and rich in protein and energy. Of these reported factors, only perceptions of nuts as healthy and rich in protein and energy were significantly associated with reported nut consumption. This may suggest that perceptions around the health benefits of nut consumption play a larger role in driving nut intake than perceptions of taste and convenience. A study among 482 consumers in China found that willingness to purchase nuts was positively associated with taste, nutrient value, hygiene, quality and safety, and brand recognition [68].

In addition to individuals’ perceptions of nut intake, the role of health professionals in promoting nut consumption must be considered. Investigation into beliefs surrounding nut consumption in lower income individuals [66] and those at risk of or diagnosed with cardiovascular disease or type 2 diabetes [67] indicated that individuals would regularly consume nuts if recommended by their doctor. Similar results were found by Yong et al. [65], with more than half of the respondents reporting that they would consume greater amounts of nuts if recommended by a doctor. However, less than 5% of respondents listed advice from their doctor as a reason they consumed nuts. In addition, only 3.8% and 2.8% reported they had been advised to eat more nuts by a dietitian and doctor, respectively. These results demonstrated a discrepancy between factors that may motivate individuals to consume nuts, and the advice received by individuals, highlighting an opportunity to encourage nut consumption through promotion by health professionals.

The role of health professionals in promoting nut intake was further explored in a cross-sectional survey of 759 dietitians, general practitioners, and practice nurses selected from the electoral roll in New Zealand [72,73]. When examined as a group, the health professionals surveyed agreed that nuts were healthy, high in protein and fat, and were filling, with approximately two-thirds of participants reporting that they recommended their patients consume more nuts [73]. However, findings differed between groups of health professionals, with dietitians found to be significantly more likely to recommend their patients eat more nuts than general practitioners or practice nurses [72]. While this finding is likely reflective of the focus on dietary advice provided by dietitians, in comparison to the broader areas of health covered by general practitioners or practice nurses, the results of this survey also indicated differences between professionals in terms of knowledge relating to nut intake. For instance, dietitians were significantly more likely to agree that nuts were healthy compared to general practitioners and nurses, with a significantly lower proportion of dietitians incorrectly believing that nut consumption increases blood cholesterol levels and increases the risk of heart disease, compared to other health professionals [73]. It is also important to note that, in the case of dietitians and general practitioners, a perception of nuts as being healthy was associated with health professionals being more likely to recommend consumption. Taken together, these findings suggest that health professionals’ own perceptions of nuts influences the degree to which they promote them to their patients, and highlights how variation in the level of knowledge of the benefits of nut intake between health professionals may influence practice. As a result, there is potential to improve knowledge of the benefits of nut consumption in health professionals (particularly non-dietetic professionals) as a strategy to encourage habitual nut intake.

## 6. Conclusions

Habitual nut intake is associated with a range of health benefits, particularly in relation to a reduction in risk of chronic diseases. As a result, food-based dietary guidelines worldwide recommend regular nut intake. However, population consumption data suggests that most individuals do not meet these recommendations. The literature has highlighted a range of barriers and facilitators to nut consumption, which should be considered when designing strategies to promote nut intake. Common barriers include confusion regarding the effects of nut consumption on body weight, the perception that nuts are high in fat, too expensive, are uncomfortable to eat for people with dental issues, and the presence of an allergy to nuts. Conversely, demographic characteristics such as higher education and income level, and a healthier lifestyle overall, are associated with higher nut intakes. An awareness of the health benefits of nut intake is associated with greater consumption, although confusion regarding the specific benefits of regularly eating nuts remains. Health professionals appear to play an important role in promoting nut intake. However, knowledge of the benefits of nut consumption could be improved across a range of health professions. Future strategies to increase nut intake to meet public health recommendations should consider clarifying misconceptions of the specific benefits of nut consumption, specifically targeting sectors of the population known to have lower nut consumption, and educating health professionals to promote nut intake.

This review provides an overview of the body of evidence relating to the barriers to and facilitators of nut consumption. It should be noted, however, that this review was not designed as a systematic review, and therefore, did not follow a predefined methodology including a set search strategy and inclusion/exclusion criteria, meaning that some relevant studies may not have been included. This review does, however, provide a broad understanding of the current evidence base on factors which may facilitate or hinder nut consumption. As a result of the relatively small body of evidence exploring these factors, and the large gap between nut recommendations and consumption levels, further research exploring these factors is justified. In particular, there is a need for research examining both facilitators and barriers in a range of populations and age groups, which may help to understand why certain characteristics are associated with higher or lower consumption of nuts. In addition, the use of more in-depth qualitative methodologies such as focus groups or interviews may improve understanding of the perceptions that underpin current consumption habits. Such research could then inform strategies to encourage higher nut consumption in line with public health guidelines.

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
