# Peer review of "Barriers and Facilitators to Nut Consumption: A Narrative Review"

_ijerph, 2020, doi:10.3390/ijerph17239127_

Round 1

Reviewer 1 Report

This review article is a fine summary of available data on nuts and possible facilitators/barriers to their consumption.

I have two major requests:

  1. To inform readers also about potential harms/hazards of nuts intake ( ie.  -potentially increased salt intake due to eating pistachios; https://www.nhs.uk/news/heart-and-lungs/pistachios-and-heart-health/ 
    - possible disastrous immunological reactions in people with peanut allergy
    - maybe others?

2. To provide also a discussion section where authors could ie. underline both pros and cons of their review (ie. non-systematical, subjective review of current evidence); what type of studies are welcomed to increase our understanding in this field.  

There are also additional comments to resolve:

  1. Please reference "core foods"; did you mean it by the main 5 food groups? (https://www.eatforhealth.gov.au/food-essentials/five-food-groups)
  2. Newer systematic reviews are available: ie. https://pubmed.ncbi.nlm.nih.gov/26548503/
  3. I suggest it should be also stated that the "positive role" of the nuts is not always confirmed: (https://bmjopen.bmj.com/content/7/11/e016863) inflammation markers; https://www.mdpi.com/1660-4601/16/24/4957/htm - body weight and blood pressure
  4. About the PREDIMED study - I suggest an addition to this summary: ... resulted in a 28% reduction in the composite risk of myocardial infarction, stroke, or death from a cardiovascular cause.
  5. [line 47] I suggest adding: "...however, the clinical significance of this reduction is negligible ("Overall, nut consumption lowers total cholesterol (TC; 0.021 to 0.30 mmol/L) and low-density lipoprotein cholesterol (LDL-C; 0.017 to 0.26 mmol/L) ref: Int. J. Environ. Res. Public Health 201916(24) 4957; 
  6. [Line 51] I suggest adding: a research tool to measure of endothelial function ref: JAMA Cardiol. 2020;5(3):360-361.
  7. [Lines 60-61]: again the negligible clinical significance of current evidence should be stated [waist circumference (WMD: − 0.51 cm, 95% CI: -0.95 to − 0.07) ref:10.1186/s12986-018-0282-y
  8. [Line 128]: Please change the formating of 11.25
  9. [Lines 103-134]: this part could be shortened, as reported by the Authors, in the most provided countries nuts consumption is far below recommended values
  10. [Lines 197-209 about allergy]: It might be both informative and interesting to readers to mention about the LEAP study conclusion: "The early introduction of peanuts significantly decreased the frequency of the development of peanut allergy among children at high risk for this allergy and modulated immune responses to peanuts." (ref: https://www.nejm.org/doi/full/10.1056/nejmoa1414850)
  11. [Lines 251-252] "While price was significantly inversely associated with the willingness to purchase nuts" - this is not a facilitator of nut consumption and should be removed from this section.

Author Response

Dear Ms Guan,

Thank you for sending the reviewer feedback on our manuscript ‘Barriers and facilitators to nut consumption: a narrative review’. We would like to thank the reviewers for their useful comments and the opportunity to clarify aspects of our manuscript. We have amended the manuscript in line with the reviewers’ comments and have outlined individual responses to the comments below.

Kind regards,

Dr Elizabeth Neale APD PhD

Career Development Fellow

School of Medicine

Faculty of Science, Medicine and Health

University of Wollongong, NSW, Australia 2522

Reviewer 1:

This review article is a fine summary of available data on nuts and possible facilitators/barriers to their consumption.

I have two major requests:

To inform readers also about potential harms/hazards of nuts intake ( ie.  -potentially increased salt intake due to eating pistachios; https://www.nhs.uk/news/heart-and-lungs/pistachios-and-heart-health/

- possible disastrous immunological reactions in people with peanut allergy

- maybe others?

Response: We have now added further information to discuss potential harms associated with nut consumption to lines 85 - 94: “Despite these benefits, there are potential disadvantages which should be considered with regards to nut consumption. Nuts may be consumed as salted varieties, and increased sodium intake is associated with increased blood pressure [37]. A small number of studies have compared the effect of different forms of nuts on health outcomes and found no significant differences in systolic and diastolic blood pressure after consumption of salted and unsalted nut varieties [38-40]. Based on these findings, Tey et al [39] concluded that there was no adverse effect on blood pressure when consuming ≤285 mg/day sodium from nuts. In addition, it should be noted that a small proportion of the population experience peanut and tree nut allergies [41-43]. For these individuals, consuming nuts results in potentially life-threatening allergic reactions, requiring the complete avoidance of the offending nut type in the diet”

  1. To provide also a discussion section where authors could ie. underline both pros and cons of their review (ie. non-systematical, subjective review of current evidence); what type of studies are welcomed to increase our understanding in this field.

Response: We have now added the following text to lines 329 - 342 (in the ‘conclusion’ section in line with the structure recommendations of the journal):

“This review provides an overview of the body of evidence relating to the barriers and facilitators to nut consumption. It should be noted, however, that this review was not designed as a systematic review, and therefore, did not follow a pre-defined methodology including a set search strategy and inclusion/exclusion criteria, meaning that some relevant studies may not have been included. This review does, however, provide a broad understanding of the current evidence base on factors which may facilitate or hinder nut consumption. As a result of the relatively small body of evidence exploring these factors, and the large gap between nut recommendations and consumption levels, further research exploring these factors is justified. In particular, there is a need for research examining both facilitators and barriers in a range of populations and age groups, which may help to understand why certain characteristics are associated with higher or lower consumption of nuts. In addition, the use of more in-depth qualitative methodologies such as focus groups or interviews may improve understanding of the perceptions that underpin current consumption habits. Such research could then inform strategies to encourage higher nut consumption in line with public health guidelines.”

There are also additional comments to resolve:

Please reference "core foods"; did you mean it by the main 5 food groups? (https://www.eatforhealth.gov.au/food-essentials/five-food-groups)

Response:  We have now clarified this statement and added references to support this point (lines 29 - 30): “Nuts (including tree nuts and peanuts) are considered to be a core food which is recommended to be consumed as part of a healthy diet [1-3].”

Newer systematic reviews are available: ie. https://pubmed.ncbi.nlm.nih.gov/26548503/

Response: We have now added Mayhew et al (2016) to the review (line 34)

I suggest it should be also stated that the "positive role" of the nuts is not always confirmed: (https://bmjopen.bmj.com/content/7/11/e016863) inflammation markers; https://www.mdpi.com/1660-4601/16/24/4957/htm - body weight and blood pressure

Response: In line with this feedback we have added a sentence to lines 60 - 61 on the evidence relating to nuts and blood pressure: “Conversely, meta-analyses suggest that nut intake does not impact blood pressure [12, 20], although there is some evidence to suggest an effect may be present for specific nut types [21].”.

With regards to the effects of nut consumption on inflammatory, we have currently referred to the inconsistent evidence demonstrated by the Neale et al (2017) meta-analysis (line 57). We have also currently commented on the lack of an effect of nuts on body weight, contrary to what would be expected given their energy dense nature, and have now included the Kim et al (2019) meta-analysis highlighted by the reviewer (line 66)

About the PREDIMED study - I suggest an addition to this summary: ... resulted in a 28% reduction in the composite risk of myocardial infarction, stroke, or death from a cardiovascular cause.

Response: Thank you, we have now amended this sentence as recommended (line 45)

[line 47] I suggest adding: "...however, the clinical significance of this reduction is negligible ("Overall, nut consumption lowers total cholesterol (TC; 0.021 to 0.30 mmol/L) and low-density lipoprotein cholesterol (LDL-C; 0.017 to 0.26 mmol/L) ref: Int. J. Environ. Res. Public Health 2019, 16(24) 4957;

Response: While some studies have found small changes in total and LDL cholesterol, as highlighted in a large meta-analysis (Del Gobbo et al, 2015) there is some evidence of dose-dependent effects, with “trials providing 100 g nuts/d lowered concentrations of LDL cholesterol by up to 35 mg/dL, an effect size comparable to some statin regimens”. As a result we feel there is evidence of a clinically significant effect particularly at higher doses. We have amended this sentence to reflect this (lines 46 - 49): “Systematic reviews of dietary interventions have also demonstrated that provision of nuts results in improvements in total cholesterol, low-density lipoprotein cholesterol, and triglycerides [12, 13], with stronger effects observed in studies assessing consumption of > 60 grams nuts/day [12].”

[Line 51] I suggest adding: a research tool to measure of endothelial function ref: JAMA Cardiol. 2020;5(3):360-361.

Response: We have now amended line 54 accordingly.

[Lines 60-61]: again the negligible clinical significance of current evidence should be stated [waist circumference (WMD: − 0.51 cm, 95% CI: -0.95 to − 0.07) ref:10.1186/s12986-018-0282-y

Response: In line with this comment, we have amended line 66 to read: “…with some evidence of small favourable effects on adiposity”

[Line 128]: Please change the formating of 11.25

Response: We have now amended the formatting of line 147.

[Lines 103-134]: this part could be shortened, as reported by the Authors, in the most provided countries nuts consumption is far below recommended values

Response: Thank you for this feedback. While we agree that the key message here is that nut consumption is far below the recommended values, we feel that the detailed information provided here gives greater context for the reader by demonstrating that intakes are low across a range of countries and by providing insights into both mean intakes and the proportion of consumers. With respect, we feel that this level of detail provides valuable insights into nut consumption patterns, which are relevant when exploring facilitators and barriers to intake.

[Lines 197-209 about allergy]: It might be both informative and interesting to readers to mention about the LEAP study conclusion: "The early introduction of peanuts significantly decreased the frequency of the development of peanut allergy among children at high risk for this allergy and modulated immune responses to peanuts." (ref: https://www.nejm.org/doi/full/10.1056/nejmoa1414850)

Response: In line with this comment, we have added the following information to lines 231 - 234 “In addition, given that research has highlighted the importance of early introduction of peanuts in infants at high risk of peanut allergies [69], further research exploring whether individuals without diagnosed allergies are unnecessarily avoiding these foods is warranted.”

[Lines 251-252] "While price was significantly inversely associated with the willingness to purchase nuts" - this is not a facilitator of nut consumption and should be removed from this section.

Response: We have now removed this sentence.

Reviewer 2 Report

Notes are found within the manuscript draft attached.

This is a review to identify the reasons people do or do not regularly consume nuts in their diet. 

There are some areas that read awkward and can be improved. Take care not to inadequately reference.

The importance of describing nut consumption habits should be more clearly emphasized.

Author Response

Dear Ms Guan,

Thank you for sending the reviewer feedback on our manuscript ‘Barriers and facilitators to nut consumption: a narrative review’. We would like to thank the reviewers for their useful comments and the opportunity to clarify aspects of our manuscript. We have amended the manuscript in line with the reviewers’ comments and have outlined individual responses to the comments below.

Kind regards,

Dr Elizabeth Neale APD PhD

Career Development Fellow

School of Medicine

Faculty of Science, Medicine and Health

University of Wollongong, NSW, Australia 2522

Reviewer 2 (extracted from comments made on pdf document for ease of reading)

Line 29: ref

Response:  We have now clarified this statement and added references to support this point (lines 29 - 30): “Nuts (including tree nuts and peanuts) are considered to be a core food which is recommended to be consumed as part of a healthy diet [1-3].”

Line 50: Sentence is awkward and lacks clarity. Are you saying, "in addition to the well-known benefits seen in coronary heart disease, there is also evidence found on certain biomarkers"? You say impact on biomarkers for a "range" of chronic conditions, but don't list them. This paragraph in particular, struggles to make your point.

Response: We have now amended lines 50 - 52 to read: “In addition to the well-known benefits of nut consumption on cardiovascular disease, there is increasing evidence of the impact of nut intake on risk factors for other health outcomes”

Line 58: You haven't actually identified a mechanism of action, you have listed the nutrient components.

Response: We have now amended lines 63 - 64 to better reflect the information contained in this paragraph and as such have removed the reference to mechanisms of action in the first sentence: “Nuts are a nutrient-dense food which are a rich source of protein, unsaturated fatty acids, fibre, and micronutrients such as folate, vitamin E, and magnesium [23, 24]”

Line 63: Your references are specific to almonds, not all nuts, which is what you allude to here.

Response: We have reworded lines 67 – 73 to read: “There are a number of mechanisms which may explain the lack of an effect of nut consumption on body weight. For instance, research conducted in almonds has shown increased excretion of fat in the faeces following almond consumption [28, 29]. This finding is thought to be due in part to the structure of the nuts, where the fat is held in plant cell walls.”

Line 65: Again, your references are to walnuts and almonds, it may be generalizable to other nuts, but there is no reference to suggest this. Please expand here to make your point more accurate

Response: Thank you for identifying this oversight (we had left out two references referring to pistachios and cashews). We have added these references and have added additional detail to clarify the specific nut varieties analysed in the studies: “As a result, the metabolisable energy available from nuts including almonds, walnuts, cashews, and pistachios is proposed to be 5 - 30% lower than estimated by Atwater factors [30-33].” (lines 73 - 75)

Line 72: This section needs to more concisely describe the benefits and therefore support your point as to why increased nut consumption would make a difference in public health.

Response: We have now added the following information to lines 83 – 85 to summarise the benefits of nut intake (prior to discussing potential risks as requested by Reviewer 1): “Habitual consumption of nuts is therefore associated with a range of health benefits particularly reductions in the risk of cardiovascular and coronary heart disease, and improvements in biomarkers such as total and low-density lipoprotein cholesterol.”

We also note that lines 115 - 124 outlines the recommendations of the Global Burden of Disease Study 2017 and Eat Lancet diet which highlight the importance of increasing nut consumption for public health benefits.

Line 74: This first paragraph seems unnecessary to make your point.

Response: In line with this feedback, we have removed the recommended text from this paragraph, with lines 96 - 103 now reading: “As a result of the health benefits associated with habitual nut consumption, nut intake is promoted in many food-based dietary guidelines globally [44]. However there is some variation in the classification of nuts and specific recommendations made”.

Line 143: It's not necessary to put n= every time, you can just write, "71 participants in a survey".

Response: We have now removed ‘n=’ when reporting the sample sizes of the studies throughout the paper.

Round 2

Reviewer 2 Report

Thank you for addressing concerns raised in the original manuscript. The clarification in the conclusion that this is not a systematic review improves the overall quality.